# Multifunctional Nanocarriers for Lung Drug Delivery

**DOI:** 10.3390/nano10020183

**Published:** 2020-01-21

**Authors:** Jorge F. Pontes, Ana Grenha

**Affiliations:** 1Centre for Marine Sciences (CCMAR), Universidade do Algarve, Campus de Gambelas, 8005-139 Faro, Portugal; pontes.jorge21@gmail.com; 2Drug Delivery Laboratory, Centre for Biomedical Research (CBMR), Universidade do Algarve, Campus de Gambelas, 8005-139 Faro, Portugal; 3Department of Chemistry and Pharmacy, Faculty of Sciences and Technology, Universidade do Algarve, Campus de Gambelas, 8005-139 Faro, Portugal

**Keywords:** antibiotics, cancer, drug delivery, lung delivery, nanocarriers, nanopharmaceuticals, proteins

## Abstract

Nanocarriers have been increasingly proposed for lung drug delivery applications. The strategy of combining the intrinsic and more general advantages of the nanostructures with specificities that improve the therapeutic outcomes of particular clinical situations is frequent. These include the surface engineering of the carriers by means of altering the material structure (i.e., chemical modifications), the addition of specific ligands so that predefined targets are reached, or even the tuning of the carrier properties to respond to specific stimuli. The devised strategies are mainly directed at three distinct areas of lung drug delivery, encompassing the delivery of proteins and protein-based materials, either for local or systemic application, the delivery of antibiotics, and the delivery of anticancer drugs—the latter two comprising local delivery approaches. This review addresses the applications of nanocarriers aimed at lung drug delivery of active biological and pharmaceutical ingredients, focusing with particular interest on nanocarriers that exhibit multifunctional properties. A final section addresses the expectations regarding the future use of nanocarriers in the area.

## 1. Introduction

The appearance of new therapies and alternative strategies for the delivery of drug molecules has been changing the paradigm of therapeutic approaches [1,2,3]. Indeed, therapeutic solutions have been implemented around one of two possible objectives: one is the development of better and more effective therapies, usually involving new drugs; the other relies on exploring different ways to deliver molecules, potentiating their action and, in many cases, simultaneously eliminating adverse effects or, at least, decreasing their impact. The latter approach has often been used for drug repurposing, finding new applications for de-risked compounds, with potentially lower overall development costs and shorter development timelines. The literature displays some recent and valuable reviews on the topic of drug repurposing [4,5]. The adverse effects derived from pharmaceuticals have always caused concern, as some can be devastating, leading to therapeutic non-compliance. Thus, exploring delivery strategies is as important as the discovery of new molecules and targets, providing molecules with specific orientation towards their targets, avoiding major biological stresses and, overall, improving the therapeutic quality [6]. Actually, in many cases, the two referred approaches are addressed simultaneously.

Considering that, in most cases, the delivery of unformulated drug molecules is not successful, formulation plays a role of utmost importance in therapy. Conventional drug delivery systems encompass numerous restrictions, which include limited targeting, low therapeutic index, poor aqueous solubility, and the potentiation of drug resistance [7]. The design and production of systems in which drug molecules are included in a carrier, being either embedded in the matrix or adsorbed to the surface, is frequently the next step towards a more effective therapy. The reasons justifying the need for drug formulation are in the annals of pharmaceutical technology, going from the simple protection of drugs to the more complex targeting of cells or tissues. In between, the need to achieve control over drug release has also been a hallmark of drug delivery research. A useful historical perspective on the generations of controlled drug delivery systems is available in [8]. In fact, in an era where drug molecules are expected to answer to increasingly complex environments, their formulation takes on a role never seen. That role assumed, a great variety of advanced drug delivery systems has been reported through the years, with important variations on their properties, including size and composition. Size is one of the most relevant features in the field. In this context, both micro- and nanocarriers have been reported to be viable approaches, the final selection being objectively dependent on the specific application that is envisaged. The potential of micron-sized carriers has been highlighted for different applications [9,10,11,12,13], but lies out of the scope of this review. As for the nanoscaled carriers, in drug delivery these fall into the designation of nanopharmaceuticals, defined by Rivera Gil et al. as “pharmaceuticals where the nanomaterial plays the pivotal therapeutic role or adds additional functionality to the previous compound” [14]. The International Organization for Standardization defines nanoparticles as those having at least one dimension less than 100 nm [15]. In turn, the American Food and Drug Administration (FDA) indicates that products involve nanotechnology, and should therefore be evaluated as such, when they are “engineered to exhibit properties or phenomena attributable to dimensions up to 1000 nm” [16]. This broader definition is the most typically seen in academic research in drug delivery and will be adopted in this review. Therefore, all submicron systems will be considered nanocarriers.

As is well indicated in the historical description of [8], after an initial period back in the 1980s and 1990s, where many micron-sized formulations became popular and reached the market, nanotechnology has been the leading interest of drug delivery scientists since the turn of the new century. In fact, the first mention to a system capable of encapsulating a molecule and providing its transport through a membrane dates back to 1965, under the name of liposome [17], which is, indeed, a nanosystem. From that point on, many other nanoformulations were described and explored, some of them consisting of particulate-based systems, with significant structural differences compared with the vesicle-based systems comprised of liposomes. Particulate carriers at the nanoscale include polymeric nanoparticles [18], solid lipid nanoparticles [19,20], nanostructured lipid carriers [21,22] and magnetic and silica nanoparticles [23,24]. Probably, issues like increased stability [6], the closer interaction with cell structures [25], the propensity to provide increased drug absorption [26] and the great ability for surface functionalization [27] have driven the higher popularity of these materials. Regrettably, the clinical translation of nanoparticulate-based systems is so far very limited, and only one formulation is available on the market: Abraxane^®^, marketed since 2005 [28]. This comprises albumin-conjugated paclitaxel, being used in metastatic breast cancer and non-small-cell lung cancer [29]. In this manner, despite the extensive research on particulate-based nanopharmaceuticals, their market absence is notorious, mainly due to tightened regulations. Even so, there are many nanoformulations currently undergoing clinical trials focusing on varied routes of administration, and it is expected that some of them make their way into the market in some years [30].

From the referred formulations undergoing a translational process and to the knowledge of the authors of this review, not one is directed to lung drug delivery. Nevertheless, this is a delivery route that has been gaining popularity in recent years, essentially owing to its non-invasiveness and the increased demonstration of its potential, not only for local therapies, but also to provide systemic action. Actually, according to the World Health Organization (WHO), chronic obstructive pulmonary disease (COPD), lower respiratory infections and lung cancer are, respectively, the third, fourth and sixth causes of death worldwide [31], which illustrates the existing therapeutic limitations. In addition, numerous other respiratory disorders are characterized by an urgent and unmet therapeutic need. The myriad of routes of administration poses the question of which one is the most adequate to deliver a drug intended to treat a specific disease. In parallel, the search for routes of administration other than the oral has been increasing for some years. The lung is now being taken into high consideration for this purpose, as clearly demonstrated in Figure 1, where the number of publications per year that specifically refer to “lung drug delivery”, as retrieved from ISI Web of Science, can be observed. It has been considered a viable alternative in the delivery of drugs and the popularity of this delivery route is reflected in the growing number of publications, especially from 2014 onwards, indicating a clear interest from the scientific community. 

The established popularity of the lung route relies on several advantages and specific features. Apart from the already mentioned ability to provide either local or systemic effect, characteristics such as high vascularization and the extensive area available for absorption are highly appealing for systemic delivery, while the low metabolic activity compared with the oral route serves both modalities [32]. Furthermore, the possibility to use lower doses and the low incidence of systemic side effects are relevant pros for local delivery [33]. A very useful and up-to-date review on the challenges and opportunities of lung delivery can be found in [33]. Despite the mentioned advantages, some limitations are also to be referred, which mainly include the mucociliary clearance as the main mechanism of defense, the patient variability on pathophysiological aspects of the organ and the need to endow the drugs with suitable aerodynamic properties to reach a specific area of the lung [34]. Regarding the latter aspect, the aerodynamic diameter of the drugs or carriers to be delivered through inhalation assumes a crucial role. The aerodynamic diameter is the diameter of a spherical particle with density of 1 g/cm^3^ and the same settling velocity as the particle of interest. In this context, it is reported that the smaller airways can be reached by particles with aerodynamic diameter lower than 5 μm, while those with less than 2 μm may arrive to the respiratory zone, which includes the alveoli [32]. 

Drugs have been administered by inhalation for millennia, but inhaled therapeutics have been predominantly used to manage common pulmonary diseases like asthma and COPD. In these areas, inhalable drugs have been dominating the market. Systemic formulations, in turn, have been facing many limitations, with significant technical hurdles requiring being addressed before success is achieved. Nevertheless, it has become consensual that, given the offered advantages, the posed challenges are worth addressing. The possibilities have long been debated, especially considering the emergence of biological drugs that are degraded in the gastrointestinal tract and, so, rely uniquely on injection to find efficacy. The scientific community has, thus, been recognizing the potential of the lung to be used as a systemic pathway, and many of the papers contributing to Figure 1 deal with systemic lung delivery, although so far this interest is not mirrored by the market. In fact, inhalable insulin is one of the exceptions to mention, appearing first as Exubera^®^, from Pfizer (2006), but being discontinued one year after approval [35,36], the company justifying the withdrawal for ‘comercial reasons’. Another product of inhalable insulin became available in 2014, as Afrezza^®^, from Mannkind Corporation, and incorporates the Technosphere^®^ technology [36,37]. These inhalable insulin products are not based on nanotechnologies, but the scientific community has been recognizing the potential of nanocarriers in lung delivery and nanoformulations have been increasingly proposed, as can be also observed in Figure 1. An integrated analysis of this figure shows that publications involving nanosystems usually comprise more than half of the total number of publications on the topic of lung delivery, which demonstrates their popularity. In fact, the superiority of nanosystems has been demonstrated in certain applications of the respiratory field, as will be described in the following sections of the review. The nanocarriers permit drug protection, provide a greater ability to interact with the tissues and cells, owing to the high surface area, often allowing specific targeting and/or controlled drug release [38]. However, the proposal of nanocarriers must not be blind, and it is important to note that some applications may take greater benefit from the use of microcarriers, for example if the therapeutic target is phagocytic cells such as macrophages. Moreover, despite the large amount of works describing nanocarriers for lung delivery applications, it is worth saying that a closer reading of the searched documents reveals that many of the works propose the nanocarriers as having potential from a conceptual point of view, but a far smaller amount approaches the practical concept of preparing the carriers for inhalation, endowing them with the required properties, namely aerodynamic, for the purpose.

While initial approaches to the development of drug nanocarriers essentially addressed issues of drug stability and control over the release, these were rapidly replaced or completed with advanced techniques of particle engineering. Thus, the proposal of more complex carriers naturally came along, with particle engineering techniques endowing the nanocarriers with specific properties well beyond their role of carrying a drug or molecule of interest. Such carriers were named as multifunctional and their applications have been explored in all areas of delivery. A multifunctional nanocarrier can be one composed of a material that provides, itself, a specific function, or one that was modified to exhibit a determined feature. Lung delivery can strongly benefit from the features of these carriers. In fact, while most asthma and COPD drugs are delivered to the lung with relatively low efficiency and still ensure therapeutic efficacy, drugs aimed at a systemic action—or used to treat orphan diseases or cancer—require the optimization of delivery efficiency. This will render the treatment cost-effective, while potentiating clinical effectiveness and minimizing side effects. 

Considering the interest of nanocarriers within the context of lung drug delivery, this review will focus on their applications, placing particular emphasis on the functionality that is provided by the proper carriers. For the effects of the review, only works addressing directly the issue of pulmonary administration of the carriers, either by adequate in vitro testing or by suitable in vivo delivery, will be considered, thus going beyond the theoretical concept of suitability for lung delivery purposes. The specific features of the carriers will be outlined and the achieved outcomes described. The envisaged applications of the nanocarriers in lung delivery are diverse, but particularly address the delivery of proteins or protein-based materials, either for local or systemic effect [39,40], cancer treatment [41,42] and local delivery of antibiotics [43,44]. Therefore, the review will specifically focus on these topics. As an introductory element to the following sections, Table 1 depicts the major respiratory diseases, along with their main limitations and the potential improvements imparted by pulmonary delivered therapy.

## 2. Lung Drug Delivery Mediated by Multifunctional Nanocarriers

Multifunctional nanocarriers can be produced from a wide range of materials. In parallel, the number of molecules that can be associated to the carriers to provide specific effects and improve their performance is also wide, either by being adsorbed or chemically bound. This variety arises from the necessity to meet different challenges and address a vast number of diseases with intrinsic different characteristics. 

In the present section, the three main topics mentioned above will be approached. As a summary of the contents, Figure 2 provides a depiction of the type of carriers used in each topic, along with the materials selected for the nanoparticle matrix and surface modification, when applicable, and also the associated molecules of interest.

### 2.1. Delivery of Proteins and Protein-Based Materials

The marked biotechnological advances observed in recent decades resulted in the appearance of many protein-based drugs. Fundamentally, the oral delivery of these molecules is prevented by the degrading effect of abundant protease content, and the possibilities of delivery are essentially reduced to injection-based strategies. However, this approach is more expensive and not appreciated by the patients, mainly due to the discomfort associated with the administration, but also because of some issues related with aesthetics, including bruising and skin marks that may compromise therapeutic compliance [45]. The pulmonary route thus appears as a sound alternative when a systemic effect is desired, but protein-based drugs also find applications in the treatment of local lung diseases. In fact, the first inhaled protein reaching the market was recombinant human DNase (rhDNase), implemented in the treatment of cystic fibrosis and available since the late 90s. Inhalable insulin appeared approximately 10 years after and lessons learnt from its development resulted in the current availability of many elegant inhalation devices and formulations. So far, no other inhaled biological drug aimed at systemic delivery has reached the market, despite those being the drugs focusing most of the attention within the context of systemic delivery mediated by inhalation. None of the referred marketed formulations encompasses the use of nanocarriers. Actually, members of our group participated in the first work proposing the inhalation of insulin encapsulated in polymeric nanoparticles, dated back to 2005 [39,46,47]. At that time, chitosan was proposed as matrix material, resulting in non-toxic nanoparticles and endowing the system with mucoadhesivity [48]. In order to provide the nanoparticles with suitable aerodynamic properties to reach the alveolar zone, a nano-in-micro system was developed, using spray-drying to microencapsulate the nanoparticles in mannitol microspheres. These, expectedly released the nanoparticles after dissolving in the lung lining fluid, providing the release of the protein that could, thus, be absorbed systemically [39]. An in vivo study in rats evidenced that microencapsulated insulin-loaded chitosan nanoparticles administered intratracheally (IT, 16.7 IU/kg) induced a more pronounced and prolonged hypoglycemic effect compared with insulin solution, as observed in Figure 3 [47], thus demonstrating the contribution of the carrier itself to the observed therapeutic effect. A similar approach was later proposed by other authors—using poly-L-lysine to modify the surface of self-assembled pure insulin nanoaggregates, benefiting from the adhesive properties of poly-L-lysine. After IT administration to diabetic rats, the modified nanostructures (5 IU/kg) induced hypoglycemic effect as stronger as subcutaneous delivery (1 IU/kg), but increasing the drug half-life from 1.28 h to 2.75 h. Although not statistically significant, the hypoglycemic effect obtained from nanoparticles was also more prolonged, achieving 23.4% relative bioavailability [49]. 

Solid lipid nanoparticles (SLN) were also proposed for this end, and were reported to provide homogeneous distribution through the lung upon delivery to diabetic rats by nebulization, showing a relative bioavailability of insulin of 22.3% comparing with subcutaneous injection [50]. An approach similar to that referred above of chitosan nanoparticles microencapsulated in mannitol microparticles was later reported for the systemic delivery of calcitonin. The inhalable carriers had mass median aerodynamic diameter (MMAD) of 2.7 μm and fine particle fraction (FPF) of 64%, the latter representing the fraction of particles with an aerodynamic diameter lower than 5 μm [51]. After IT administration, around 85% relative bioavailability was determined, compared with subcutaneous delivery. The bioavailability was also superior to that obtained after the inhalation of native calcitonin [52]. Another approach in the same line, proposed the delivery of IgG mediated by poly(lactic-*co*-glycolide) acid (PLGA)-based nanoparticles produced by double emulsification and subsequently spray-dried to acquire suitable aerodynamic properties. Leucine was further included to improve aerosolisation. MMAD around 4 μm and FPF of approximately 50% indicated suitability to reach the deep lung, while a prolonged release of up to 35 days was observed in PBS pH 7.4, enabling applications where prolonged release is envisaged [53].

More than simply avoiding injections, the driving force fostering investment on systemic drug delivery through the lung relies on the improvement of pharmacokinetics, which could be an advantage for drugs currently delivered through the oral, buccal or transdermal routes. The studies reported above reinforce the potential of the lung to provide an access to the systemic compartment, but above all, they show that the nanocarriers can play a role in improving the therapeutic performance. Nevertheless, one of the limitations that is relatively tranversal to works on lung delivery is the fact that, in most cases, the in vivo testing of inhalable formulations is performed by IT administration, which does not mimic the reality when human delivery is concerned. This aspect still requires some advancement in order to better predict in vivo outcomes.

The delivery of drugs that are specifically directed to the lung is the other side of the picture of lung delivery. Local treatment of lung diseases usually aims at low systemic bioavailability in order to avoid the risk of unwanted side effects in other organs due to rapid drug translocation via the air–blood barrier. Some respiratory pathologies are ineffectively treated with existing small molecule-based therapies. RNAi effectors, such as small interfering RNA (siRNA), have been shown to enable the post-transcriptional silencing of key molecular disease factors that cannot be readily targeted with conventional small molecule drugs [54]. Therefore, some therapeutic alternatives are currently being proposed in this context. The type of cell that is targeted in this approach is variable and depends on the specific airway disease. Epithelial cells are key players in cystic fibrosis, for instance, while dendritic cells, macrophages and T lymphocytes are targets in inflammatory diseases like asthma or COPD [54]. The local therapeutic response to siRNA can be markedly enhanced through the use of nanoparticles, essentially due to the possibility to provide specific cell targeting. The period of time that siRNA is retained in the lung plays an important role on the success of the approaches. This period is affected by rapid elimination due to mucociliary clearance, translocation to systemic circulation and secondary organs, and phagocytosis by alveolar macrophages. The complexation of siRNA with polyethylenimine (PEI), forming polyplexes, was demonstrated to reduce the translocation and extend siRNA retention time in lung, while preventing substantial phagocytosis by macrophages and avoiding extensive mucociliary clearance [55]. Additionally, it has been shown on several occasions [56,57] that the contact of nanoparticles with the surfactant present in the alveolar zone leads to the coating of nanocarriers by a biomolecular corona, composed of lipids and proteins. This corona affects nanoparticle hydrophobicity and possibly enhances biorecognition, with consequences on the subsequent interactions with cells and other biological entities. Most works report a negative impact of this process on the therapeutic outcomes. Interestingly, with regards to the delivery of siRNA, recent works have suggested that modifying the surface of siRNA-loaded nanoparticles with lung surfactant (by a simple incubation) provides improved siRNA transfer activity due to facilitated cellular uptake [54]. Improved transfection efficiency of pDNA was also reported previously in presence of lung surfactant [58]. siRNA-dendrimer (polyamidoamine, generation 4) complexes of ~100–130 nm were microencapsulated in trehalose-inulin microparticles, which displayed aerodynamic diameters of 4.5–5.5 μm, adequate to reach the deep lung. These microparticles dissolved in aqueous medium, releasing the nanocomplexes, which showed enhanced cellular uptake and transfection in RAW264.7 macrophages, compared with native siRNA [59]. Protein-based molecules are, in many cases, regarded as sensitive and their manipulation in delivery devices such as inhalers is often feared. A study demonstrated the stability of mRNA upon nebulization, showing no effect of nebulization on protein duration of action or the cytotoxicity of the formed PEI polyplexes [60].

Inhalable vaccines have also been the focus of several works and, although many pulmonary vaccines have been proposed, only few involve nanocarriers. An interesting approach was reported that uses a double emulsion formed by water/PLGA in organic/lactose-water, with IgG, the model antibody, dissolved in the inner aqueous phase. The emulsion was spray-dried, resulting in PLGA nanoparticles within lactose microparticles. Suitable properties for inhalation were observed, with 60% FPF. Submicron-particles were released after contact with aqueous medium, and approximately 70% IgG released after 6 days in pH 7.4 [53]. Another approach reported poly(glycerol adipate-co-ω-pentadecalactone) (PGA-co-PDL) nanoparticles that were modified to express on their surface the pneumococcal surface protein A, which is an important antigen of *S Pneumoniae* (~20 mg antigen/mg of nanoparticles). Nanoparticles of approximately 150 nm were then microencapsulated in leucine microparticles to provide respirability. The latter registered an MMAD of 1.7 μm and a 74% FPF, which grants the ability to reach the broncho-alveolar zone, potentiating the uptake by dendritic cells, as has been demonstrated experimentally [61]. Silica nanoparticles were also reported for this end. Nanoparticles were associated with plant-derived H1N1 influenza hemagglutinin antigen (HAC1), and proposed as an inhalable vaccine against the influenza virus. A mucosal adjuvant (bis-(3′,5′)-cyclic dimeric guanosine monophos-phate (c-di-GMP)) was further tested. After IT vaccination of mice, the double-adjuvanted vaccines (nanoparticles plus mucosal adjuvant) were observed to induce high systemic antibody responses, comparable to the systemic vaccination control. Moreover, local IgG and IgA responses were observed in the bronchoalveolar lavage [62].

The described works clearly demonstrate that the lung provides a suitable route for the delivery of protein-based molecules, serving, in this context, the purpose of both systemic and local delivery.

### 2.2. Delivery of Antibiotics

The delivery of antibiotics to the lung seems a very reasonable approach in the treatment of infections that are based in that organ. In fact, the most common routes of delivery of antibiotics are the oral and parenteral, even if the treatment of respiratory infections is intended. Addressing local lung infections requires reaching effective concentrations of the drug in the organ, which implies the administration of significantly high doses and a general exposure of the organism to the drugs. The direct administration to the infection site would, thus, permit using lower doses and avoid or decrease systemic exposure, with the consequent reduction in systemic side-effects. Additionally, the more targeted delivery is a premise to decrease the incidence of antimicrobial resistance, an important current goal in antibiotic therapy [63,64]. Antibiotic resistance has been, for many years, one of the greatest public health problems. The increasing misuse of these molecules, ever since their discovery, has been making bacteria progressively resistant, by means of the development of specific cellular mechanisms. This has been continuously and consistently posing a renewed challenge to the treatment of infectious diseases [65]. 

The market makes available some formulations of inhaled antibiotics, including tobramycin, colistin and aztreonan, which are mainly directed to the treatment of infections associated with cystic fibrosis conditions [66]. Other applications have been reported occasionally, such as the use of aerosolized antibiotics in hospital-acquired pneumonia [67]. Research in the area has been increasing consistently, and a recent review on inhalable antibiotic formulations is available in [66]. Along with the discovery of new antibiotics, the development of delivery systems to improve the therapeutic performance of the molecules has been object of scientific efforts and both approaches are, in fact, effective countermeasures against antibiotic resistance. The search for new drug molecules is known to be slower than the development of the drug delivery systems that lead the antibiotics to the intended site of action. Of the marketed formulations referred above, not one is based on nanocarriers, but the literature provides many works reporting their use to improve the performance of lung-delivered antibiotics, addressing, among others, the improvement of kinetic profiles and issues related to side effects. One of the most common respiratory infections is tuberculosis, caused by *Mycobacterium tuberculosis*, which primarily accumulates and replicates inside alveolar macrophages located in the alveolar zone of the lung [68]. Despite the existence of effective therapy of tuberculosis for many decades, the fact is that it still remains a global epidemic, being a major healthcare problem, as portrayed by the last data published by WHO [69]. Not only is the established therapy prolonged and associated with severe side effects—which decrease therapeutic compliance—but also, the issues of co-morbidity with HIV and the existing bacterial resistance are relevant. A great number of works propose the use of nanocarriers for tuberculosis treatment, in many cases envisaging lung delivery applications. Very frequently, the developed carriers involve strategies of surface chemical functionalization, namely mannosylation. The rationale behind this approach is based on the fact that bacterial hosts, the macrophages, have several surface receptors that are likely to be used as therapeutic targets [70,71]. The mannose receptor is one of the main, which may provide a favorable interaction with some units and chemical groups present on the carriers’ surface, including the mannose units, but also others like fucose and *N*-acetylglucosamine [72]. In principle, considering that the bacteria are hosted by the macrophages located in the alveoli, this is the zone to be reached in the design of any strategy aimed at treating tuberculosis by lung delivery. 

SLN have been proposed as carriers for this end. Rifabutin-loaded SLN prepared with glyceryl dibehenate (~100 nm) were further encapsulated in mannitol microparticles to acquire adequate aerodynamic properties to reach the alveolar zone (~44% of particles with less than 6.4 μm). An in vivo test in a murine model of infection (*Mycobacterium tuberculosis* strain H37Rv) demonstrated that the inhalation of the dry powder permitted the effective delivery of the antibiotic to the lung, along with drug distribution to liver and spleen. Moreover, an enhancement of antibacterial activity was observed compared to nontreated animals [43]. Another formulation of SLN, this time composed of palmitic acid and cholesteryl myristate and loaded with rifampicin (~400 nm), was further freeze-dried to obtain an inhalable powder. An MMAD of around 5–7 μm and an FPF within 30% and 50% were determined. The SLN were mannosylated to improve their targeting ability, which was verified experimentally, with increased macrophage uptake (~80%) compared to non-functionalized SLN (~40%) [71,73]. Rifampicin was also the chosen antitubercular drug to encapsulate in polymer-glycerosomes, which showed to be more stable than conventional liposomes [74]. These are phospholipid/glycerol vesicles combined with trimethyl chitosan or hyaluronic acid (80–110 nm). Upon nebulization, an MMAD of approximately 4 µm was obtained along with an FPF of up to 77%. In any case, the aerodynamic performance of the carriers was always better than that of the free drug, and drug incorporation in the vesicles was found to increase its efficacy against *Staphylococcus aureus*. Following IT administration to rats, glycerosomes promoted the accumulation of rifampicin in the lung, with lower systemic distribution, and low accumulation in other organs. The formulation containing hyaluronic acid was found to perform more favorably [75]. Although it was not discussed, the use of hyaluronic acid might be beneficial due to a favorable interaction of its *N*-acteylglucosamine units with CD44 [76,77] and mannose receptors [78]. Chitosan and chitosan-folate were further used to functionalize oleic acid-based nanoemulsions loaded with rifampicin, which were nebulized to render adequate respirability (MMAD of 3–4 μm and FPF of 62–73%). It was found that chitosan-folate provided increased cell internalization, proposed to result from a favorable interaction with macrophages by both chitosan units and folate groups. Additionally, this formulation provided in vivo higher lung drug content and reduced plasma drug concentration [79]. Chitosan nanoparticles prepared by ionic gelation with tripolyphosphate were also proposed a couple of times as carriers in antitubercular drug delivery. A first work described the association of isoniazid and used spray-drying with lactose and leucine to reach an FPF of 45% [80]. More recently, similar nanoparticles were associated with bedaquiline (size varying within 70 and 700 nm depending on preparation conditions). A powder form of the nanoparticles was obtained by freeze-drying, registering 28% FPF and 3.38 μm MMAD—which was better than the conventional DPI formulation used as control (15% FPF and MMAD of 4 μm). The study determined the absence of toxicity of the nanoparticles in vivo in rats and further demonstrated a higher drug concentration in lungs upon inhalation of the microencapsulated nanocarriers [81]. Frequently, the choice of chitosan as nanoparticle matrix material is not explicitly justified, leaving the readers with the sensation that the polymer is only used because of its high popularity, a natural consequence of its favorable properties regarding mucoadhesion and absence of toxicity. In this latter work, the authors justified the positive results with a possible favored uptake of nanoparticles by alveolar macrophages mediated by an interaction of chitosan positive charges (from amino groups) with the negatively charged surface of macrophages. However, most of the works fail to point out that the strong affinity of macrophages to chitosan is possibly a result of the recognition of *N*-acetylglucosamine units of the polymer by macrophage surface receptors, as was proposed in a work from our group reporting chitosan microparticles as antitubercular drug carriers [78,82]. The use of chitosan as a matrix material was also proposed in genipin-crosslinked carboxymetylchitosan nanoparticles loaded with isoniazid and rifampicin, which were freeze-dried to obtain a powder. After inhalation by rats, a greater accumulation of drugs was observed in the lung upon the delivery of the carriers compared with free drugs. Additionally, extended residence time of drugs in the lung was achieved and lower levels in other organs (liver, kidney) were registered [83]. 

As a whole, several nanoparticle-based formulations are proposed in the frame of tuberculosis therapy, in most cases showing improved results attributed to specific functionalization of their surface or benefits from their proper composition (e.g., chitosan). In order to provide adequate respirability, the nanocarries are either nebulized or transformed in inhalable powders using spray- or freeze-drying. In the works showing in vivo results, the delivery by inhalation typically provided increased lung concentrations of the drug and lower systemic exposure.

Other lung diseases work as a door for opportunistic infections, cystic fibrosis being a major example. This is a genetic disorder caused by mutations in the cystic fibrosis transmembrane conductance regulator (CFTR) gene. This gene is of the utmost importance, as it encodes a protein that forms an ion channel in epithelial cell membranes. The genetic dysfunction may translate into different defects of the protein, in any case ending up in bronchial obstruction that occurs due to the secretion and accumulation of a thick and sticky mucus in the airways. The accumulation of mucus creates the adequate conditions for bacterial colonization, which typically involves *Pseudomonas aeruginosa* and *Staphilococcus aureus* [84,85,86]. This justifies that cystic fibrosis therapy requires regular administration of antibiotics, apart from bronchodilators and mucolytics. 

A solution of tobramycin for inhalation was the first approved aerosolized antibiotic to be used against *P. aeruginosa* and, recently, a dry powder form of tobramycin has become available. However, this drug shows poor mucus penetration, rapid clearance and suboptimal concentrations at the site of infection, which are frequently not enough to stop the complications derived from the bacterial infection [87]. The need for better therapies is one of the emergent objectives in the field of cystic fibrosis. Nanotechnology can bring forth some solutions in this context. Mucus penetration is, indeed, a major issue. If it is possible to overcome this barrier, enabling a more effective delivery of drugs, infections can be eliminated with higher efficiency. In a very interesting work, Schneider et al. (2017) demonstrated that mucus-penetrating nanoparticles (polystyrene nanoparticles coated with polyethylene glycol—PEG) of sizes up to 300 nm have higher retention in the lung and more uniform distribution compared with similar sized nanoparticles devoid of PEG and, thus, mucoadhesive [88]. Regrettably, no biological assays were reported so far, either in vitro or in vivo. Colistin was encapsulated in PLGA nanoparticles which were further surface-modified with chitosan (270 nm) or polyvinyl alcohol (PVA, 330 nm) and then spray-dried to reach adequate aerodynamic properties. An MMAD less than < 5 μm was obtained when lactose was used as carrier, while the use of mannitol resulted in MMAD < 8 μm. In vitro assays revealed increased ability of chitosan-modified particles to penetrate artificial mucus and also suggested a role of the nanoparticles in potentiating the anti-biofilm activity of colistin, possibly due to the ability of nanoparticles to penetrate the biofilm and to sustain drug release [89]. A previous work from the same group, where tobramycin was encapsulated in similar nanoparticles, demonstrated in vivo that PVA-modified nanoparticles reached the alveoli, while particles modified with chitosan tend to appear in the upper airways, possibly as a consequence of their specific aerodynamic characteristics [90]. Ciprofloxacin was self-assembled with PEG-*g*-phthaloyl chitosan nanoparticles (218 nm) and further microencapsulated by spray-drying in swellable alginate microparticles (volume mean diameter of 3.9 μm). Upon IT delivery to rats, the encapsulated molecule was found in higher concentrations in lung tissue and lung lavage compared with the administration of the control consisting of a physical mixture of lactose and micronized drug [91]. 

Importantly, many bacteria regulate pathogenicity via a cell-to-cell communication system that is known as quorum sensing. This is dependent on cell density and involves the production of virulence factors to coordinate group behaviors [92]. Antibacterial strategies based on the inhibition of quorum sensing are currently growing and this represents, indeed, a novel form of therapy. A very interesting approach in the delivery of antibiotics for the treatment of *Pseudomonas aeruginosa* infection involved SLN (<100 nm) loaded with a quorum sensing inhibitor. Nebulization has resulted in an MMAD of 2.2 μm, and an FPF of around 85% was determined, enabling the deposition of a certain fraction in the bronchial region. The SLN demonstrated to penetrate into artificial sputum, but the most important finding was that the proper SLN have anti-virulent effect, acting in addition to the quorum sensing inhibitor to decrease the virulence factor pyocyanin [93]. 

As can be verified, under the scope of antibiotic delivery, a great deal of attention is given to tuberculosis. However, the number of works addressing antibiotic delivery mediated by nanocarriers that present a certain degree of multifunctionality while providing real demonstration of potential for lung delivery represents only a fraction of the literature. Apart from tuberculosis, *Pseudomonas aeruginosa* and *Staphilococcus aureus* are the two main targets, being frequently associated with cystic fibrosis and pneumonia, although they can be also involved in hospital-acquired lung infections, for instance. It was demonstrated in the several works described that the nanocarriers can provide extra strength to antibiotic-mediated therapies.

### 2.3. Applications in Cancer Therapy

The WHO refers to lung cancer as one of the most lethal cancers [94]. In 2018, 18.4% of cancer-related deaths were a result of lung cancer, and the number of new cases (11.8%) was one of the highest, on par with breast cancer [95]. WHO has a set of goals to fight cancer aggressively, and the development of new strategies in cancer treatment is a worldwide priority. Lung cancer can be categorized into non-small cell lung cancer (NSCLC) and small cell lung cancer (SCLC). NSCLC is considered aggressive and comprises approximately 85% of all occurrences, in which various subtypes are included, such as adenocarninoma and squamous cell lung cancer. SCLC is even more aggressive, comprising the remaining 15% of cases [96]. Both the high probability of metastasis derived from SCLC and the frequently late diagnosis, contribute to the high mortality [97,98]. At earlier stages, the treatment for both types of lung cancer is surgery, enabling the removal of the affected area. However, at later stages, chemotherapy and radiation are the valid options, often to reduce the tumor mass before any surgical procedure [97,99]. Regrettably, these options have a great impact on patient’s physiology, as both cancerous and healthy cells are attacked, resulting in symptoms that are difficult to manage. As a consequence, patient susceptibility to other diseases is increased. 

The scientific community has been working to develop more targeted therapies, which is facilitated by the increasing information on molecular pathways, specific receptors and the cancer microenvironment, enabling different treatment approaches. Although the intravenous route is the most frequently used to deliver anticancer drugs, including in lung cancer, the use of the lung route is an alternative yet to be fully explored in lung cancer therapy. This approach would allow a more targeted delivery, directly reaching the affected area, possibly with higher effectiveness than that provided by systemic delivery. Importantly, the lung can be considered the main route for the delivery of anticancer drugs in cases of lung cancer, but can also be used as add-on therapy for the treatment of lung metastasis secondary to other cancers. Overall, it is considered that this approach would potentially enable the use of lower doses of anticancer drugs, with reduced systemic exposure and consequent residual metabolization of the molecules [100]. This strategy further helps in the reduction in adverse effects, contributing to the increased quality of life of the patients.

The number of nanocarriers proposed for an application in lung cancer mediated by lung delivery is high. In most cases, a therapeutic effect is envisaged, but some of the works address diagnostic purposes. Although this is of great importance in cancer, especially at early stages of development, these strategies will not be detailed further, as they are out of the scope of the review. For further reading on this matter, Silva et al. (2019) and Mottaghitalab et al. (2019) comprise two comprehensive reviews on potential diagnostic strategies [101,102]. Therefore, only works on nanocarriers envisaging therapeutic approaches will be considered.

The general observation of the literature indicates that, in most cases, the proposal of nanocarriers for an application in cancer therapy implies functionalization; that is, carriers with some sort of surface modification that benefits their interaction with the tumor environment. One of the strategies often reported in this context relies on the use of a matrix that is added of molecules potentially recognized by cell receptors prevailing in cancer cells comparing with healthy cells. Such an approach was already discussed briefly in the previous section, referring to carriers endowed with cell-targeting ability mediated by mannose moieties. In the context of lung cancer, lactoferrin-chondroitin sulfate nanocomplexes (~190 nm) were reported to co-deliver doxorrubicin (Dox) and elagic acid. The latter was first converted into water soluble nanocrystals due to its hydrophobicity. The nanocomplexes were prepared by electrostatic interaction between lactoferrin and chondroitin sulfate and the two drugs incorporated during this process. Due to the overexpression of CD44 and lactoferrin receptors on the surface of lung cancer cells, these nanocomplexes were shown to have favored cell recognition, mediated by chondroitin sulfate and lactoferrin content, respectively. The authors further hypothesized that clathrin-mediated endocytosis could have contributed favorably to the internalization of nanocomplexes, as their size is within the range of the pore size of the clathrin receptor (up to 200 nm) [103]. Therefore, the functionality of these carriers is provided not only by their size but also by their composition, which ensures specific targeting ability. To provide adequate aerodynamics for lung delivery, the nanocomplexes were then microencapsulated into a mannitol matrix, reaching FPF close to 90% and MMAD of 2.56 μm. After IT insufflation of the microencapsulated nanocomplexes in tumor-bearing mice, tumor growth biomarkers were quantified and revealed lower levels when the inhalable formulation was used, in comparison with the inhalation of free drugs or intravenous administration [42].

These cell recognition strategies were also addressed in works with gold nanoparticles. Such carriers have strong interest in cancer therapy, finding applications in photothermal therapy, radiotherapy and also as drug carriers. Their inhalation has been demonstrated to provide lung accumulation, which can be useful in lung cancer therapy [104]. A very recent review on the topic is available in [105]. Gold nanoparticles (2 nm) that were coated with functional derivatives of thiolated PEG have shown invisibility towards the immune system provided by PEG [106], but also enabled attaching other moieties to provide specific targeting. The surface of the nanocarriers was thus modified with the ligand RGD—a peptide with a relatively high and specific affinity for the integrins overexpressed in tumor neovasculature [107,108]. A mice model of single-nodule lung adenocarcinoma [109] was used to establish which route of administration, either inhalation or intravenous delivery, would be more effective on adenocarcinoma targeting using the nanocarriers. The biodistribution data demonstrated higher concentration of the carriers upon inhalation [110]. In another approach, gold nanoparticles were loaded with temozolomide (~40 nm)—an alkylating agent already in use in other cancer types. The IT administration to healthy mice indicated the safety of gold nanoparticles upon quantification of lactate dehydrogenase and the tumor markers carcinoembryonic antigen and alpha-fetoprotein. The proper carriers were reported to induce oxidative damage and ability to inhibit cell proliferation and cell cycle in G1-phase, while the delivery of drug-loaded carriers to mice bearing lung cancer demonstrated a synergic effect between the carriers and the loaded drug [111].

The optimization of the interaction of nanocarriers with cancer cells has also been reported using SLN. A complex nanodelivery system based on SLN was proposed, being composed of multi-compartmental lipid nanocomposites (190–225 nm). Berberin and rapamycin, with demonstrated a synergic anticancer effect, were initially encapsulated in SLN. To optimize the rate of delivery of both drugs, multicompartment systems were developed. Berberin was incorporated as hydrophobic ion pair with sodium dodecyl sulfate in SLN’s core, sustaining its release, while rapamycin was pre-formulated as phospholipid complex, thus helping to improve its solubility and relatively enhance its release. The tumor-targeting ability was improved by layer-by-layer assembly of the cationic lactoferrin and the anionic hyaluronic acid, which target the CD44 and lactoferrin receptors overexpressed by lung cancer cells. Adequate aerodynamics were achieved after spray-drying with a mixture of mannitol/maltodextrin/leucine (MMAD of 3.3 μm, FPF of 56%). An assay in mice bearing lung tumors demonstrated that inhaled nanocomposites induced a decrease in lung weight compared with the inhalation of free drugs, along with reduction in tumor size and levels of angiogenic markers [112]. Another work proposed the modification of the SLN surface with a chitosan derivative that was previously added of folate moieties [113]. The authors hypothesized that both the chitosan derivative and the folate engraftments would increase the retention of the nanoparticles within the lungs, and activate the folate receptors, increasing the amount of drug delivered to cancer cells. The nanocarriers (~250 nm; +32 mV) provided slower release of paclitaxel after coating (58% in 3 days) and demonstrated binding affinity to cell lines expressing the folate receptor. In in vivo assays, a higher lung paclitaxel concentration was observed for the inhaled chitosan-coated SLN compared with the intravenous administration of the drug. Moreover, drug concentration was higher at 1 h and 6 h post-administration for the coated formulation compared with inhaled and intravenous delivered paclitaxel. As a final remark for this study, the authors noticed that the SLN were distributed throughout the solid lung tumors, with low interaction with the vessels, which occurs with systemic delivery of anticancer agents. Paclitaxel was also loaded in PEG-polylactic acid (PLA) nanoparticles that were further conjugated with the epithelial cell adhesion molecule (EpCAM, CD326), which was also overexpressed in lung cancer. IT delivery of nanoparticles to c-Raf transgenic lung cancer mice permitted reducing drug toxicity, with animal surviving increasing from 20% to 70% [114]. Another approach proposed lipid polymeric nanoparticles (hydrophobic polymeric core, phospholipid layer and an outer layer of epidermal growth factor (EGF), PEG and distearoylphosphoethanolamine) targeting the EGF receptor (EGFR) [115], which is overexpressed in lung carcinoma [116,117]. Cisplatin and Dox were the associated drugs. The presence of EGF in the outer part of the nanoparticle promoted the interaction with EGFR, leading to the release of drugs at the cancer site. An in vivo assay revealed tumor inhibition ratio of ~75%.

Chitosan-coated PLGA nanoplexes were proposed to carry an antisense oligonucleotide against the human telomerase RNA component, as telomerase activity is detected in most NSCLC. The potential of the oligonucleotide as a telomerase inhibitor has been described [118], although its poor cellular uptake hinders its use in cancer therapy. The nanoplexes (±160 nm) were delivered IT to healthy mice, using the model of the isolated perfused and ventilated lung, and provided increased uptake of the oligonucleotide by the epithelium than that observed for the free form of the oligonucleotide. Although no specific study was performed, the authors justified the results with the potential ability of nanoparticles to escape pulmonary clearance mechanisms [119].

A solution towards a resistant form of cancer was proposed as inhalable self-assembled nanoparticles comprised of human serum albumin (HSA), tumor necrosis factor (TNF)-related apoptosis-inducing ligand (TRAIL) and Dox [120]. The latter was conjugated to HSA and formed nanoparticles, which were then coated with TRAIL (342 nm). Initial tests in H226 cells, which are representative of NSCLC, have shown that the simultaneous presence of Dox and TRAIL enabled increased cytotoxic potential, as cell viability after 3 days of exposure decreased from approximately 60% when only one of the molecules was present in HSA nanoparticles, to 20%–30% after dual association. An in vivo assay was then performed in lung tumor-bearing mice, delivering the nanoparticles in the form of micron-sized liquid droplets. The tumors of mice treated with HSA nanoparticles combining TRAIL and Dox were much smaller and lighter than those of mice treated with the corresponding nanoparticles containing only one of the molecules, TRAIL or Dox. Haloperidol was also used as ligand to enhance targeting ability of albumin-based nanoparticles (218 nm). Nanoparticles were prepared via the desolvation of bovine serum albumin, previously conjugated with haloperidol and loaded with Dox. Spray-drying with mannitol, trehalose and leucine produced nano-in-microparticles with aerodynamic diameter of 4.6 μm and an FPF of 66% [121].

Some works further report therapeutic approaches that rely on the ability of the carrier matrix to respond to different stimuli [122,123]. The modification of temperature and pH are the usual stimuli to be used [124], setting the basis for the elaboration of the so called *smart polymers* or *systems*. The rationale behind their use is that if a certain stimulus (pH value or temperature) is reached, it will trigger a phase transition in the carrier matrix, leading to the release of the drug in a predetermined site. In this context, a copolymer based on methoxy poly(ethylene glycol)-poly(ethylenimine)-poly(L-glutamate) was produced and nanoparticles prepared (<75 nm), using electrostatic interaction and chelate effect to encapsulate simultaneously Dox and cisplatin [125]. In vitro assays demonstrated increased release of Dox at acidic pH, showing the capacity to release the drug in a cancer setting. The pulmonary administration of the nanocarriers to mice with metastatic lung cancer was performed using a Microsprayer aeroliser, resulting in the increased accumulation of carriers within the lungs, along with low concentrations in other tissues, especially in the area surrounding tumor lesions. It was hypothesized that the smaller size of the carriers benefitted their penetration in the cancer mass, while an ineffective vessel arrangement prevented a systemic dissemination. The results also shown decreased tumor masses, suggesting increased efficacy of the nanoformulation. Dendrimers of poly(amidoamine) were also used in a similar pH stimulation strategy. Dox was conjugated with the polymer and the dendrimers were spray-dried with mannitol to endow suitable aerodynamic characteristics (FPF was 40%–60%). Dendrimers readily released from microparticles in aqueous medium and drug release was only found to occur in response to intracellular pH drop [126]. In another work, similar dendrimers showed strong time-dependent toxicity in Calu-3 cells, a model of the respiratory epithelium, which was attributed to sustained drug release. It was also shown that the conjugation of PEG molecules to the dendrimer improved their permeation across the cell layer, in a concentration-dependent manner. In this case, the dendrimers were formulated in a pressurized metered dose inhaler, leading to aerosols with 82% FPF and MMAD of 1.3 μm [127]. Dendrimers of PEGylated polylysine, also conjugated with Dox, were IT administered to a syngenic rat model of lung metastised breast cancer. An administration twice a week led to an over 95% reduction in lung tumors after two weeks, compared with the IV administration of Dox solution, which resulted in a reduction of 30%–50% [128]. 

In some cases, a combination of the above-mentioned strategies is proposed, as happens in a work reporting a stimuli-responsive core-shell nanoparticle conjugated with folic acid. The rationale behind this formulation was to create a pH- and temperature-sensitive network, comprised by a copolymer of poly(*N*-isopropylacrylamide) and carboxymethylchitosan, which comprises the shell of the nanosystem. In turn, the core is comprised of PLGA and an image contrast agent (superparamagnetic iron oxide, SPIO). While PLGA allows the controlled release of the encapsulated molecules, gemcitabine in this case, SPIO serves the dual role of contrasting agent and inductor of temperature change by external application of an alternating magnetic field. SPIO-induced temperature alterations lead to conformational change of polymeric shell, allowing drug release. Additionally, the shell of the system is pH-sensitive, providing drug release at the acidic pH characteristic of cancer environment. Moreover, the delivery is even more targeted owing to the surface conjugation with folic acid, benefitting from the overexpression of the folate receptor in cancer cells [129]. The nanocarriers (~289 nm, −36 mV) evidenced increased cell uptake in the presence of a magnet, as a consequence of the presence of SPIO in the formulation. In vivo assays in lung tumor-bearing mice showed decreased tumor volume compared with the controls. The pulmonary retention of nanoparticles was confirmed by magnetic resonance imaging (MRI) and, when coupled with radiotherapy, a synergic effect takes place to slow tumor growth. 

Finally, magnetic nanoparticles have also been proposed several times within the scope of lung cancer. Many reports explore an application in lung cancer diagnosis, using Fe_3_O_4_ paramagnetic cores [130] or gadolinium-based particles [131,132]—the latter further enabling a radiosensitising effect. Nevertheless, therapeutic actions are also proposed. Iron oxide nanoparticles (Fe_3_O_4_; 56 nm, −49 mV) were spray-dried with lactose and doxorubicin, reaching an MMAD of 3.27 μm. An in vitro study demonstrated that—compared with a liquid suspension—the microencapsulated nanoparticles provided more than twice the deposition and retention of particles in regions under the influence of a strong magnetic gradient [133].

As a whole, several different strategies are described that end up with positive results in lung cancer treatment. Nevertheless, cancer research still has much ground to cover, and the associated therapeutics are growing at the rate of deciphering of new receptors and new molecular cascades. Nanotechnology is progressing with these discoveries, to provide improved strategies for cancer treatment. Those described above are dominated by the optimization of the carriers surface, either by engineering with specific ligands, by carefully selecting the matrix components or by combining all the effects, in order to provide more targeted delivery of the drugs and an intimate contact with cancer cells, which will thus result in improved therapeutics. 

## 3. Expectations for the Future

Pharmaceutical technology has been playing a vital role on medicine, as it allows us to explore different materials and their combinations to prepare drug carriers, and to further endow these with better properties that enable us to reach the desired target sites, ending up with therapeutic success. Many questions arise around the topic of nanocarrier-based lung drug delivery. Scientists have given many and varied answers in attempt to address all the rising issues, finding alternatives and engineering adequate systems to fulfill requirements and needs. One of the concerns is always the fate of the drug. In lung drug delivery, the objective is sometimes to retain the drug in the lung—as happens in local delivery approaches, thus minimizing the systemic absorption. In other cases, a systemic effect is the desired outcome and the carriers are engineered to avoid retention. The options to address therapeutic demands are varied, as seen by the plethora of systems, alternatives and engineering possibilities described through the review. As expected, the global analysis reveals that most of the works focus on the use of the lung route to attain local rather than systemic effects. It was also verified that, from the three main topics explored in the review, the delivery of anticancer drugs is among the most prominent focuses in the literature, which is justified by the severity of the numbers associated with this disease, the increasing number of patients, and the lack of therapeutic options, besides the marketing appeal of cancer therapies. 

Whichever the specific topic, it has become clear that the use of nanocarriers is an added value and may provide an evolution of therapeutic responses if used properly and the arising toxicological concerns are addressed. Furthermore, the engineering of different strategies mainly involved the surface functionalization of the carriers—or at least took benefit of their components to provide specific effects. The latter strategy perhaps still requires further investment to reveal its potential, as *smart polymers* have emerged as a new range of powerful tools, but still need refinement. The use of different physiological conditions, such as pH, temperature and redox compounds, or even light at different wavelengths, can be the answer to more targeted and efficient therapies. Objectively, the field needs clinically feasible formulations, which possibly could combine some of the different strategies that were described, certainly using some kind of surface engineering to reach specific biological targets, but also adjusting the desired properties with the use of materials that may respond specifically, either to stimuli or to the established characteristics of the target area. Recently, a very interesting and promising approach was reported in this context, comprising a nanoparticle-in-microgel system that provides drug release triggered by the presence of proteases. As the presence of these enzymes is greatly increased in the lung as a consequence of the inflammatory processes related to asthma, COPD and cystic fibrosis, this may comprise a therapeutic strategy for the treatment of these conditions [134]. Despite the tremendous advancement, the evolution of the field is strongly dependent on new knowledge being generated in more basic science; namely, the molecular mechanisms of the diseases, which are great indicators of tools to be used in the development of new therapies. Additionally, it cannot be forgotten that many of the tools identified as successful and providing improved therapeutic responses—that is, carriers and above all, the materials—are so far not approved by the regulatory entities for an application in lung delivery. This poses a great challenge itself. Addressing the toxicity of inhaled therapeutic nanocarriers is matter of inescapable importance. For many years now it has become clear that the biocompatibility of nanomaterials is not that of the raw materials and its evaluation needs to go much beyond the assessment of the isolated components. The nanomaterial must be considered a new entity instead, within the context of a specific delivery route [135]. Therefore, generating data on the safety of the nanocarriers and the new materials identified as potential adjuvants, in the framework of the lung route, is currently understood as an urgent need to potentiate lung drug delivery applications. This should involve toxicity tests that evaluate all the possible toxicity pathways, both in vitro and in vivo, while ensuring that the 3Rs policy to reduce, refine and replace the use of animals in research is followed. The initial in vitro tests should address cytotoxicity and genotoxicity, and should also evaluate potential epigenetic toxicity [136]. The fate of the proper carriers after the delivery is often disregarded. A very recent study comparing the clearance kinetics of liposomes and solid lipid nanoparticles after IT delivery of suspensions to rats has shown similar clearance rates, despite different deposition patterns [137]. Studies around this topic are, thus, imperative to provide data on the safety of the materials and the kick-off to their clinical application. 

It is important to point out that, when in vivo assays were described, which occurred in a considerable number of the presented works, IT delivery of the nanocarriers was the predominant technique for the assessment, which implies a high risk when establishing possible correlations with human delivery. The fact that nanocarriers themselves do not exhibit suitable aerodynamics for inhalation is also relevant, as this always implies an extra step, typically proposed to involve the spray-drying of nanocarriers to produce nano-in-microcarriers that can deposit in the lung.

The area still needs to evolve in several topics before inhalable nanocarriers enter clinical trials. Not only the question of performing more realistic in vivo assays is determinant, but also the toxicological assessment plays a defining role. Helpful technologies have been arising, such as 3D printing, which was used on the printing of artificial airways that enabled the study of particle flowability and dose assessment. Lim et al. described this application in the neonate, showing a powerful tool to improve the ethics associated with formulation testing and to provide solutions for children born with respiratory complications [138].

All in all, this review highlighted an integrative process that considers progress made at the level of basic science, which clarifies pathophysiological aspects of each clinical condition, and the development of tools and strategies to reach the pharmacological targets. Many works were described with inhalable nanocarriers that have displayed potential, even with the existing limitations. All the issues, however, point to a common objective of providing the knowledge to enable the engineering of nanocarriers that will promote improved lung therapeutics.

## Figures and Tables

**Figure 1 nanomaterials-10-00183-f001:**
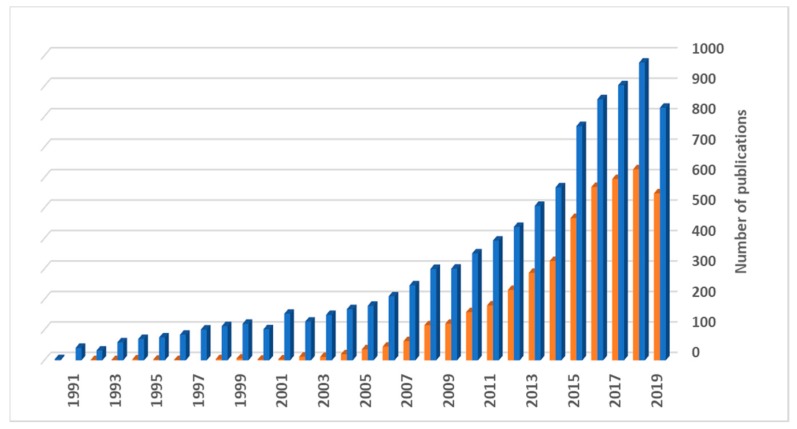
Number of scientific publications under the topics of “lung drug delivery” (blue) and “lung drug delivery and nano” (orange) on ISI Web of Science, as function of the publication year (last updated in January 2020).

**Figure 2 nanomaterials-10-00183-f002:**
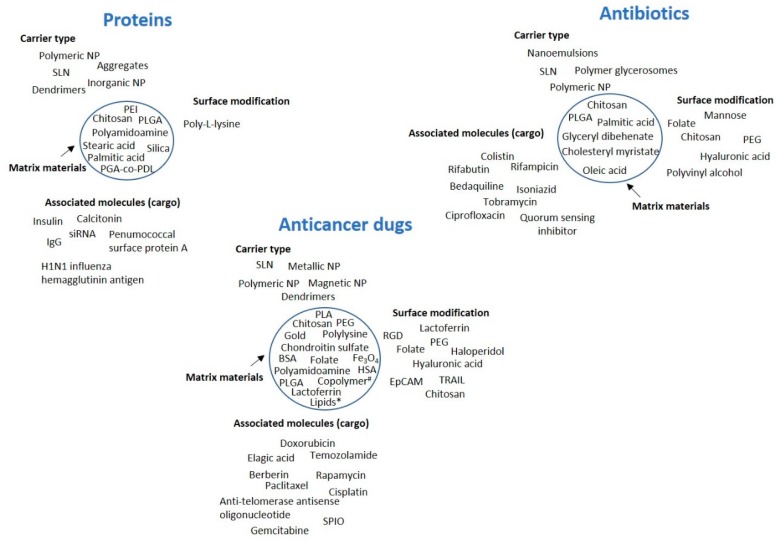
General overview of the types of nanocarriers used in the delivery of proteins, antibiotics and anticancer drugs, along with the materials applied in the carrier matrix (inside the circle), the ligands used for surface functionalization and the associated molecules of interest (the cargo). The circle indicates the carrier. BSA: bovine serum albumin, EpCAM: epithelial cell adhesion molecule, HSA: human serum albumin, IgG: immunoglobulin G, NP: nanoparticles, PEG: polyethylene glycol, PEI: polyamidoamine, PGA-co-PDL: poly(glycerol adipate-co-ω-pentadecalactone), PLA: polylactic acid, PLGA: polylactic-co-glycolic acid, RGD: tripeptide Arg-Gly-Asp, siRNA: small interfering RNA, SLN: solid lipid nanoparticles, SPIO: superparamagnetic iron oxide, TRAIL: tumor necrosis factor-related apoptosis-inducing ligand, ^#^ copolymer based on methoxy poly(ethylene glycol)-poly(ethylenimine)-poly(L-glutamate), * lipids: glyceryl monostearate, cholesterol.

**Figure 3 nanomaterials-10-00183-f003:**
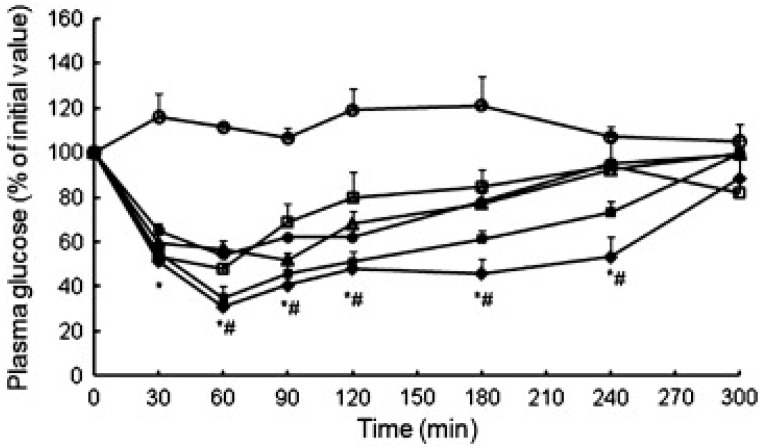
Hypoglycemic profiles following intratracheal administration to rats of microencapsulated insulin-loaded chitosan nanoparticles (INS-loaded CS NPs) prepared using chitosans of different MW (CS 113 and CS 213), and control formulations (mean ± SD, *n* ≥ 3): (◊) Microencapsulated INS-loaded CS NPs—CS 113; (▪) Microencapsulated INS-loaded CS NPs—CS 213; (◦) Microencapsulated blank (without insulin) CS NPs—CS 113; (□) Mannitol microspheres containing INS; (Δ) Suspension of INS-loaded CS NPs—CS 113; (●) INS solution in PBS pH 7.4; * Statistically significant differences from microencapsulated blank CS NPs (*p* < 0.05); # Statistically significant differences from INS solution (*p* < 0.05). Reprinted with permission from [47].

**Table 1 nanomaterials-10-00183-t001:** General overview of the major respiratory diseases, along with their main limitations and improvements imparted by pulmonary delivery of the drugs. Indication of the application, in each disease, of the drug classes addressed in the review.

Respiratory Disease	Main Limitations	Improvements from Lung Delivery	Proteins	Antibiotics	Anticancer Drugs
Asthma	Low therapeutic efficacy of delivered drugs; inefficient control of the disease; airway inflammation	n.a. *	x		
COPD	Persistent inflammation; parenchymal lung tissue destruction; abnormalities of the small airways	n.a. *	x	x	
Pneumonia	Low amount of drug reaches infection site; antimicrobial resistance	Higher drug accumulation in infection site; co-localisation of drug and infectious agent	x	x	
Cystic fibrosis	Thick viscous mucus; recurrent lung infections; progressive impairment of lung airways	Mucus-penetrating carriers; increased lung drug retention; delivery of genetic material to restore CFTR function	x	x	
Tuberculosis	Reduced amount of drug reaches infection site; antimicrobial resistance; long therapeutic regimen	Co-localisation of drug and infectious agent; reduction in antibiotic resistance incidence; possibility of add-on therapy (along with oral); reduce treatment duration		x	
Lung cancer	Non-specificity of drugs; difficulties to reach the affected tissues; severity of systemic adverse effects	Vectorisation to cancer cells; reduction in systemic adverse effects			x

CFTR: Cystic fibrosis transmembrane conductance regulator; COPD: chronic obstructive pulmonary disease. n.a.: not applicable; * conventional therapy already administered via inhalation.

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
