# Peer review of "Multifunctional Nanocarriers for Lung Drug Delivery"

_nanomaterials, 2020, doi:10.3390/nano10020183_

Round 1

Reviewer 1 Report

Tha review manuscript entitled "Multifunctional nanocarriers for lung drug delivery" by Pontes and Grenha, represents an interesting state-of-the-art on the different strategies recently proposed to exploit the pulmonary route.

The manuscript is well organized and well-written, however some typing errors are present and the manuscript requires a revision made from an English mother-tongue.

However, it is not clear the time window taken in consideration by the Authors. By figures 1 and 2, it seems they will discuss the topic from its very beginning to recent publications, which sounds a difficoult goal to achieve.

Actually, going over the paper, it seems that Authors focus on papers published from 2004, but they only refer to one manuscript per year...We suggest to limit the time window, thus having the possibility to better deepen articles published in the last 5 or at least 10 years. Indeed, in the Reviewer's opinion, some very important recent studies on SLN and polymeric NP are still missing, whose discussion in mandatory in the optic to release an update review on the topic.

In the legends of figures 1 and 2, please indicate the last update, at least month/year.

Author Response

The review manuscript entitled "Multifunctional nanocarriers for lung drug delivery" by Pontes and Grenha, represents an interesting state-of-the-art on the different strategies recently proposed to exploit the pulmonary route.

The manuscript is well organized and well-written, however some typing errors are present and the manuscript requires a revision made from an English mother-tongue.

The authors appreciate the comments from the Reviewer. The English style was carefully revised before submission of the revised version.

However, it is not clear the time window taken in consideration by the Authors. By figures 1 and 2, it seems they will discuss the topic from its very beginning to recent publications, which sounds a difficult goal to achieve. Actually, going over the paper, it seems that Authors focus on papers published from 2004, but they only refer to one manuscript per year...We suggest to limit the time window, thus having the possibility to better deepen articles published in the last 5 or at least 10 years. Indeed, in the Reviewer's opinion, some very important recent studies on SLN and polymeric NP are still missing, whose discussion in mandatory in the optic to release an update review on the topic.

Figures 1 and 2 were designed to show the readers the progression of the area of lung drug delivery since its very beginning. The figures are also a broad spectrum, not focusing solely on the three main molecule types that were later defined by us for a more detailed approach (protein-based molecules, antibiotics, anticancer drugs). Furthermore, we have restricted the works to be approached to those addressing realistically the delivery of the carriers to the lungs, either by aerodynamic assessment or in vivo assays involving lung delivery.

Nevertheless, following the advice of the reviewer we have checked our search on database. We found a couple of other works from late 2019 on polymeric nanoparticles that comply with the requirements to be described in detail in our review. No works were found on SLN. In fact, there are other articles on SLN or polymeric nanoparticles but either they do not reach an inhalation driven approach (at least in vitro aerodynamic assessment) or, very frequently, the carriers show no multifunctionality. Actually, this was a difficulty also for the authors, as in many cases the described works were very interesting but were lacking the multifunctionality that was the driving force of the review.

In the legends of figures 1 and 2, please indicate the last update, at least month/year.

Figures 1 and 2 were combined into a unique figure following the advice of another reviewer. The date was included as advised and, as 2019 has now reached its end, we have updated the graphic to include the year 2019.

Reviewer 2 Report

The article “Multifunctional nanocarriers for lung drug delivery” by Jorge F. Pontes et al. is dedicated to review of protein-, antibiotics- and anticancer drug-based materials modified by diverse methods to act as nanocarriers for special purposes. The review presented by authors is important and covers a large area of knowledge related to the topic. In my opinion, the article should be published in MDPI Nanomaterials Journal, but a certain revision of the article is needed as outlined below:

282-284 lines. It is written, that lung delivery compared to injection could have an advantage by way of improvement of pharmacokinetic. However, here and in general in the whole text there are no pointed disadvantages related to injection-based methods. It would be well to supplement the review by this information as far as the authors compare the injection to lung delivery. For instance, besides pharmacological imperfection of injection-based methods, there are disadvantages related to esthetics (marks on the skin, bruises, in some cases an appearance of skin ulcers)

308-313 lines. It is written: “…the contact of nanoparticles with the surfactant present in the alveolar zone leads to the coating of nanocarriers by a biomolecular corona, composed of lipids and proteins. This corona affects nanoparticle hydrophobicity and possibly enhances biorecognition, with consequences on the subsequent interactions with cells and other biological entities”. Could the author clarify, whether there are works related to protein-based nanoparticles already covered by alveolar surfactants before using them as nanocarriers? And do any particles have an ability to be covered by these surfactants? What characteristics should particles have for this? The authors should elucidate this issue.

Among the article, the authors use the phrase “adequate aerodynamic properties”. Could the authors elucidate the meaning, and perhaps, quantitative characteristics of values describing these properties except particles size?

The article is called “Multifunctional nanocarriers for lung drug delivery. There are several cited articles in the text where the size of the smallest functional part of a particle equaled to several hundred nanometers (for instance, line 394) and more. The authors have to clarify, whether we can attribute these particles as nanocarriers. Are they submicron particles, perhaps? Where is a boundary for lung delivery particles below of which nanoscale size imparts them special properties other than submicron and micron particles. The authors should elucidate this issue.

Line 704. There is a typo in the word “knew”.

Author Response

The article “Multifunctional nanocarriers for lung drug delivery” by Jorge F. Pontes et al. is dedicated to review of protein-, antibiotics- and anticancer drug-based materials modified by diverse methods to act as nanocarriers for special purposes. The review presented by authors is important and covers a large area of knowledge related to the topic. In my opinion, the article should be published in MDPI Nanomaterials Journal, but a certain revision of the article is needed as outlined below:

282-284 lines. It is written, that lung delivery compared to injection could have an advantage by way of improvement of pharmacokinetic. However, here and in general in the whole text there are no pointed disadvantages related to injection-based methods. It would be well to supplement the review by this information as far as the authors compare the injection to lung delivery. For instance, besides pharmacological imperfection of injection-based methods, there are disadvantages related to esthetics (marks on the skin, bruises, in some cases an appearance of skin ulcers)

The authors thank the reviewer for the comments and specifically for highlighting this point. The raised issue was addressed and the text revised accordingly (new manuscript version, page 7, line 225-228).

308-313 lines. It is written: “…the contact of nanoparticles with the surfactant present in the alveolar zone leads to the coating of nanocarriers by a biomolecular corona, composed of lipids and proteins. This corona affects nanoparticle hydrophobicity and possibly enhances biorecognition, with consequences on the subsequent interactions with cells and other biological entities”. Could the author clarify, whether there are works related to protein-based nanoparticles already covered by alveolar surfactants before using them as nanocarriers? And do any particles have an ability to be covered by these surfactants? What characteristics should particles have for this? The authors should elucidate this issue.

The referred sentence intends to transmit that, after lung delivery, nanoparticles will contact with the lung surfactant, promoting an interaction that ends up in the formation of a biomolecular corona. This, in turn, will affect all the subsequent interactions. The sentence written after that one (Interestingly, with regards to the delivery of siRNA, recent works have suggested that the simultaneous formulation of siRNA and surfactant in nanocarriers may provide benefits [48]) indicates, in fact, the existence of works reporting that modifying the nanoparticle surface with pulmonary surfactant (AlveofactÒ) during their preparation, led to improved siRNA transfer activity due to facilitated cellular uptake. The assay, which finds parallel in other works, simply involves an incubation of nanoparticles with lung surfactant for a predetermined period of time (4h).

With the comment of the reviewer we realised that the text is perhaps not very clear. Therefore, we performed some alterations in the sentences envisaging clarification of the exposed ideas. The sentences now read: “Interestingly, with regards to the delivery of siRNA, recent works have suggested that modifying the surface of siRNA-loaded nanoparticles with lung surfactant (by a simple incubation) provides improved siRNA transfer activity due to facilitated cellular uptake [54]. Improved transfection efficiency of pDNA was also reported previously in presence of lung surfactant [58].”. (new manuscript version, page 8, lines 322-326).

Among the article, the authors use the phrase “adequate aerodynamic properties”. Could the authors elucidate the meaning, and perhaps, quantitative characteristics of values describing these properties except particles size?

We thank the reviewer for this comment. There is real need to include this indication and we apologise for not noticing its absence before. A new sentence was included in the revised version of the manuscript (page 3, lines 132 – 137), reading “(…) the aerodynamic diameter of the drugs or carriers to be delivered through inhalation assumes a crucial role. The aerodynamic diameter is the diameter of a spherical particle with density of 1 g/cm3 and the same settling velocity as the particle of interest. In this context, it is reported that the smaller airways can be reached by particles with aerodynamic diameter lower than 5 μm, while those with less than 2 μm may arrive to the respiratory zone, which includes the alveoli [32]”. We have further detailed the definition of Fine Particle Fraction (FPF), which is also frequently mentioned through the text in the description of the works (new manuscript version, page 8, line 282).

The article is called “Multifunctional nanocarriers for lung drug delivery”. There are several cited articles in the text where the size of the smallest functional part of a particle equaled to several hundred nanometers (for instance, line 394) and more. The authors have to clarify, whether we can attribute these particles as nanocarriers. Are they submicron particles, perhaps? Where is a boundary for lung delivery particles below of which nanoscale size imparts them special properties other than submicron and micron particles. The authors should elucidate this issue.

The reviewer is right that we should have defined the object of the review regarding the size criterium of eligibility. We have added a sentence reading “The International Organization for Standardization defines nanoparticles as those having at least one dimension less than 100 nm [15]. In turn, the American Food and Drug Administration (FDA) indicates that products involve nanotechnology, and should therefore be evaluated as such, when they are “engineered to exhibit properties or phenomena attributable to dimensions up to 1000 nm” [16]. This broader definition is the most typically seen in academic research in drug delivery and will be adopted in this review. Therefore, all submicron systems will be considered nanocarriers.” (new manuscript version, page 2, lines 62-68).

Line 704. There is a typo in the word “knew”.

This was corrected.

Reviewer 3 Report

The authors have reviewed several nano sized carriers and their applicability in drug delivery to the lungs. The review is well written and in sufficient detail and several important literature pertaining to the topic has been covered (including the most recent ones) and discussed in detail. However there are some suggestions that could be worked upon for more organization and ease of grasping the main perspective of the authors address through this review:

The introduction is descriptive and provides an overview of the various types of lung anomalies that the nanocarriers are being equipped to address. Here the authors have highlighted at several places how nanocarriers with multiple functionalizations can be used for specific types of problems. However for the ease of reading, I suggest the authors can exclusively highlight in one of the initial paragraphs as to what are some of the most common and intriguing clinical problems faced in the lung drug delivery (E.g in terms of delivery route or complexity of the target tissues or size of the blood vessels etc) and regarding why multifunctional nanocarriers are actually advantageous over conventionally used (or currently used) treatments.

Figures 1 and 2 can be combined into one. Moreover since 'nano' is a subsection under 'lung drug delivery it can be shown in the same graph with sub-bars (for 'nano') with main bars (for lung drug delivery.

More information on what are the challenges and recent nanomaterial engineering developments to address chronic lung obstruction and inflammation could be provided. It would also be great to have a summarized table based on major types of lung ailments and columns with:  where the three classes (shown in figure 3) fit in, what are the major limitations and most recent improvements that have gone in these directions.    

The data shown in figure 3 is slightly difficult to read and interpret. It is not clear what class of materials are indicated inside the circle or if they also belong to nanocarriers. Does the associated molecules mean the drug/agent i.e the cargo being carried? It would be rather interesting to have it in the form of a branch/tree diagram list or as a flowchart list with each category i.e proteins, anticancer drugs and antibiotics being the main columns. (this could be based on the data that the authors have already reviewed)

Perhaps in the last chapter, Authors can also highlight from their own viewpoint regarding why there has been so many past studies and substantial investment with different classes of nanocarriers with so many variables such as particle size, functional groups and drug release profiles and what could formulate towards a highly clinically feasible formulation or technology considering the key challenges involved in lung drug delivery. The authors have already discussed about fate, toxicity and delivery of the drug. Perhaps they can discuss more on what set of nanocarriers and functionalities have not worked so far in all three categories. This information can help as a hint and inspire the material researchers to work towards (or not to work towards) certain directions and avoid duplicity of studies 

The language and grammar is good overall, I suggest another round of proofreading to correct some typos , E.g: line 704: ‘knew’, Line 690: ‘objetive’

Author Response

The authors have reviewed several nano sized carriers and their applicability in drug delivery to the lungs. The review is well written and in sufficient detail and several important literature pertaining to the topic has been covered (including the most recent ones) and discussed in detail.

We thank the reviewer for this comment.

However there are some suggestions that could be worked upon for more organization and ease of grasping the main perspective of the authors address through this review: The introduction is descriptive and provides an overview of the various types of lung anomalies that the nanocarriers are being equipped to address. Here the authors have highlighted at several places how nanocarriers with multiple functionalizations can be used for specific types of problems. However for the ease of reading, I suggest the authors can exclusively highlight in one of the initial paragraphs as to what are some of the most common and intriguing clinical problems faced in the lung drug delivery (E.g in terms of delivery route or complexity of the target tissues or size of the blood vessels etc) and regarding why multifunctional nanocarriers are actually advantageous over conventionally used (or currently used) treatments.

In fact, the conventional use of the lung route pertains essentially to unformulated anti-inflammatory drugs and antibiotics, not associated with carriers. The use of delivery systems and, specifically in this case, of the multifunctional nanocarriers, is proposed to either improve the performance of some existing drugs or to permit the use of drugs which do not show satisfactory therapeutic efficacy in the absence of the carrier. In many cases, the carrier is the element facilitating the delivery or the needed interaction with the specific structures of the organism and this was already explicit in the last paragraph of the introduction.

The observation of the reviewer is, however, pertinent and, therefore, the manuscript was updated to express the mentioned issues. Two paragraphs were included in the introduction, one reading “Despite the mentioned advantages, some limitations are also to be referred, which mainly include the mucociliary clearance as the main mechanism of defence, the patient variability on pathophysiological aspects of the organ and the need to endow the drugs with suitable aerodynamic properties to reach a specific area of the lung [34]. Regarding the latter aspect, the aerodynamic diameter of the drugs or carriers to be delivered through inhalation assumes a crucial role. The aerodynamic diameter is the diameter of a spherical particle with density of 1 g/cm3 and the same settling velocity as the particle of interest. In this context, it is reported that the smaller airways can be reached by particles with aerodynamic diameter lower than 5 μm, while those with less than 2 μm may arrive to the respiratory zone, which includes the alveoli [32].”(new manuscript version, page 3, lines 128-137). The other paragraph reads “. In fact, the superiority of nanosystems has been demonstrated in certain applications of the respiratory field, as will be described in the following sections of the review. The nanocarriers permit drug protection, provide a greater ability to interact with the tissues and cells, owing to the high surface area, often allowing specific targeting and/or controlled drug release [38]. However, the proposal of nanocarriers must not be blind and it is important to note that some applications may take greater benefit from the use of microcarriers, for example if the therapeutic target is phagocytic cells such as macrophages.” (new manuscript version, page 4, lines 156-162).

Figures 1 and 2 can be combined into one. Moreover since 'nano' is a subsection under 'lung drug delivery it can be shown in the same graph with sub-bars (for 'nano') with main bars (for lung drug delivery.

Following the advice of the reviewer, we joined both figures in a sole figure.

More information on what are the challenges and recent nanomaterial engineering developments to address chronic lung obstruction and inflammation could be provided. It would also be great to have a summarized table based on major types of lung ailments and columns with:  where the three classes (shown in figure 3) fit in, what are the major limitations and most recent improvements that have gone in these directions.   

A table was included that considers the proposal of the reviewer (page 5).

The data shown in figure 3 is slightly difficult to read and interpret. It is not clear what class of materials are indicated inside the circle or if they also belong to nanocarriers. Does the associated molecules mean the drug/agent i.e the cargo being carried? It would be rather interesting to have it in the form of a branch/tree diagram list or as a flowchart list with each category i.e proteins, anticancer drugs and antibiotics being the main columns. (this could be based on the data that the authors have already reviewed)

The authors thank the reviewer for this comment and the suggestion. We have made the effort to elaborate a flowchart as advised, but we did not like the result, as the chart is too crowded. With all due respect for the opinion of the reviewer, we have thus decided to maintain the initial figure. However, as the comment of the reviewer evidences lack of clarity of the figure, we have improved it in order to transmit a clearer message.

In fact, with “associated molecules” we mean the cargo, as is explained in the figure caption, and they can be either encapsulated or adsorbed to the surface of the carrier. In turn, the materials inside the circle are the matrix materials, while the circle itself was the way to indicate the carrier.  We added “matrix materials” close to the central circle and improved the caption, to make it clearer to the readers.

Perhaps in the last chapter, Authors can also highlight from their own viewpoint regarding why there has been so many past studies and substantial investment with different classes of nanocarriers with so many variables such as particle size, functional groups and drug release profiles and what could formulate towards a highly clinically feasible formulation or technology considering the key challenges involved in lung drug delivery. The authors have already discussed about fate, toxicity and delivery of the drug. Perhaps they can discuss more on what set of nanocarriers and functionalities have not worked so far in all three categories. This information can help as a hint and inspire the material researchers to work towards (or not to work towards) certain directions and avoid duplicity of studies

The strategies and works described in the review cover a lot of ground and, specially, a lot of different variables (materials, sizes, charges…), making a direct answer on which set of nanocarriers would be more adequate, a difficult task. However, we understand the intention of the author and, so, the last section was improved to better reflect our opinion. Please consult from line 724 on (new manuscript version, page 16).

The language and grammar is good overall, I suggest another round of proofreading to correct some typos, E.g: line 704: ‘knew’, Line 690: ‘objetive’

The authors thank the reviewer for this comment. The manuscript was revised again to eliminate the existing typos.

Reviewer 4 Report

In this review, the authors introduced the popular nanomaterials that were designed for lung drug delivery. Three categories of drugs were discussed in detail, i.e. protein-based drugs, antibiotics, and cancer treatment drugs. Even though the author did a very comprehensive summary of the published work, the structure of this manuscript needs to be improved to better deliver the idea. Please see the comments below.

In the introduction, the authors didn’t cover the major challenges associated with lung drug delivery. More importantly, they didn’t explain the reason why nanomaterials will be better than other forms of drug. The current paragraphing is confusing. The terms, such as proteins, antibiotics and cancer drug are not exclusive to each other. I guess a better way is to arrange the paragraphs based on the type of nanomaterials and summarize the desired properties that best solve challenges in lung drug delivery. Figure 3 could be reorganized based on comment 2. It might be worth to discuss the toxicity of nanomaterials and the prevention in the future outlook.

Author Response

In this review, the authors introduced the popular nanomaterials that were designed for lung drug delivery. Three categories of drugs were discussed in detail, i.e. protein-based drugs, antibiotics, and cancer treatment drugs. Even though the author did a very comprehensive summary of the published work, the structure of this manuscript needs to be improved to better deliver the idea. Please see the comments below.

In the introduction, the authors didn’t cover the major challenges associated with lung drug delivery. More importantly, they didn’t explain the reason why nanomaterials will be better than other forms of drug.

We thank the reviewer for the comments. The observation of the reviewer is pertinent and, therefore, the manuscript was updated to address the mentioned issues. Two paragraphs were included in the introduction, one reading ““Despite the mentioned advantages, some limitations are also to be referred, which mainly include the mucociliary clearance as the main mechanism of defence, the patient variability on pathophysiological aspects of the organ and the need to endow the drugs with suitable aerodynamic properties to reach a specific area of the lung [34]. Regarding the latter aspect, the aerodynamic diameter of the drugs or carriers to be delivered through inhalation assumes a crucial role. The aerodynamic diameter is the diameter of a spherical particle with density of 1 g/cm3 and the same settling velocity as the particle of interest. In this context, it is reported that the smaller airways can be reached by particles with aerodynamic diameter lower than 5 μm, while those with less than 2 μm may arrive to the respiratory zone, which includes the alveoli [32].”(new manuscript version, page 3, lines 128-137). The other paragraph reads “. In fact, the superiority of nanosystems has been demonstrated in certain applications of the respiratory field, as will be described in the following sections of the review. The nanocarriers permit drug protection, provide a greater ability to interact with the tissues and cells, owing to the high surface area, often allowing specific targeting and/or controlled drug release [38]. However, the proposal of nanocarriers must not be blind and it is important to note that some applications may take greater benefit from the use of microcarriers, for example if the therapeutic target is phagocytic cells such as macrophages.” (new manuscript version, page 4, lines 156-162).

The current paragraphing is confusing. The terms, such as proteins, antibiotics and cancer drug are not exclusive to each other. I guess a better way is to arrange the paragraphs based on the type of nanomaterials and summarize the desired properties that best solve challenges in lung drug delivery. Figure 3 could be reorganized based on comment 2.

We did the effort of arranging the review in a different way, but we were not happy with the final result. With all due respect for the opinion of the reviewer, we have decided to maintain its original structure. Figure 3 (figure 2 in the revised version) was however a bit modified to be clearer, having now a more detailed description.

 It might be worth to discuss the toxicity of nanomaterials and the prevention in the future outlook.

The toxicity of nanomaterials is indeed a very relevant issue and, recognising that, we had already addressed the question in the original version of the manuscript. Nevertheless, we have now extended the comments on nanotoxicology in the referred section, adding extra text that reads “For many years now it has become clear that the biocompatibility of nanomaterials is not that of the raw materials and its evaluation needs to go much beyond the assessment of the isolated components. The nanomaterial must be considered a new entity instead, within the context of a specific delivery route [132]. Therefore, generating data on the safety of the nanocarriers and the new materials identified as potential adjuvants, in the framework of the lung route, is currently understood as an urgent need to potentiate lung drug delivery applications. This should involve toxicity tests that evaluate all the possible toxicity pathways, both in vitro and in vivo, while ensuring that the 3Rs policy to reduce, refine and replace the use of animals in research is followed. The initial in vitro tests should address cytotoxicity and genotoxicity, and should also evaluate potential epigenetic toxicity [133].” (new manuscript version, page 17, lines 743-752).

Round 2

Reviewer 1 Report

The manuscript was revised by the Authors, according to the Reviewers' suggestion, therefor in the present form the papre can be published in Nnomaterials.

Author Response

We thank the reviewer evaluation.

Reviewer 4 Report

The authors have made a great effort to address the concerns I have. However, I still found the figure 2 is not well organized. Maybe the author can turn it into a table to specify the class of drug, cargo, carrier, application and reference. 

Author Response

We stongly apologise but, although we respect the opinion of the reviewer, we do not consider it is a better option to replace the figure by a table or a diagram. We consider that a figure as it is, is more appealing for the readers. The table becomes more descriptive, "boring", and it is not really our intention. We consider it does not benefit the work. As none of the other 3 reviewers considered the figure misleading, we propose that the figure is included.